# L-Methionine Protects against Oxidative Stress and Mitochondrial Dysfunction in an In Vitro Model of Parkinson’s Disease

**DOI:** 10.3390/antiox10091467

**Published:** 2021-09-15

**Authors:** Mariano Catanesi, Laura Brandolini, Michele d’Angelo, Elisabetta Benedetti, Maria Grazia Tupone, Margherita Alfonsetti, Enrico Cabri, Daniela Iaconis, Maddalena Fratelli, Annamaria Cimini, Vanessa Castelli, Marcello Allegretti

**Affiliations:** 1Department of Life, Health and Environmental Sciences, University of L’Aquila, 67100 L’Aquila, Italy; mcatanesi@unite.it (M.C.); michele.dangelo@univaq.it (M.d.); elisabetta.benedetti@univaq.it (E.B.); mariagrazia.tupone@univaq.it (M.G.T.); margherita.alfonsetti@guest.univaq.it (M.A.); annamaria.cimini@univaq.it (A.C.); 2Dompé Farmaceutici SpA, Via Campo di Pile, 67100 L’Aquila, Italy; laura.brandolini@dompe.com (L.B.); daniela.iaconis@dompe.com (D.I.); 3Pharmacogenomics Unit, Istituto di Ricerche Farmacologiche Mario Negri IRCCS, 20156 Milan, Italy; enrico.cabri@marionegri.it (E.C.); maddalena@marionegri.it (M.F.); 4Center for Biotechnology, Sbarro Institute for Cancer Research and Molecular Medicine, Temple University, Philadelphia, PA 19122, USA

**Keywords:** nutraceuticals, neurodegeneration, oxidative stress, mitochondria, brain, reactive oxidative species, dietary supplement

## Abstract

Methionine is an aliphatic, sulfur-containing, essential amino acid that has been demonstrated to have crucial roles in metabolism, innate immunity, and activation of endogenous antioxidant enzymes, including methionine sulfoxide reductase A/B and the biosynthesis of glutathione to counteract oxidative stress. Still, methionine restriction avoids altered methionine/transmethylation metabolism, thus reducing DNA damage and possibly avoiding neurodegenerative processes. In this study, we wanted to study the preventive effects of methionine in counteracting 6-hydroxydopamine (6-OHDA)-induced injury. In particular, we analyzed the protective effects of the amino acid L-methionine in an in vitro model of Parkinson’s disease and dissected the underlying mechanisms compared to the known antioxidant taurine to gain insights into the potential of methionine treatment in slowing the progression of the disease by maintaining mitochondrial functionality. In addition, to ascribe the effects of methionine on mitochondria and oxidative stress, methionine sulfoxide was used in place of methionine. The data obtained suggested that an L-methionine-enriched diet could be beneficial during aging to protect neurons from oxidative imbalance and mitochondrial dysfunction, thus preventing the progression of neurodegenerative processes.

## 1. Introduction

Parkinson’s disease (PD) is the second most frequent neurodegenerative disorder, with prevalence increasing with age. The molecular mechanism of PD is still undefined, but one important factor in the disease pathogenesis is oxidative stress [1,2], which can stimulate the oxidation of cell constituents, comprising amino acids. Among them, methionine is oxidized to methionine sulfoxide (Met-O) by adding an oxygen atom to its reactive sulfur atom, and the reduction reaction is catalyzed by the family of methionine sulfoxide reductases (MSRs) [3]. It has been suggested that the reversible oxidation and reduction cycle relating to MSRs aim in counteracting the deleterious effects of oxidative stress [3,4] and are determining factors of lifespan [5,6].

High levels of reactive oxygen species (ROS) detected in PD are due to the mitochondrial impairment leading to protein oxidation and aggregation as well as apoptosis [7]. Unfortunately, there is no cure for PD, and there is an urgent need to find a therapy since the elderly population and, thus, neurodegenerative disorders are significantly increasing. A potential approach to counteract the disease onset or the progression could be the use of nutraceuticals or antioxidants.

Indeed, nutraceuticals and antioxidants are presently receiving attention as adjuvants in PD treatment, and several investigations using both in vitro and animal models of this disease are published. The antioxidant dietary supplement can help to prevent extreme oxidative stress environments by limiting ROS production and counteracting ROS-related damage.

In particular, in this research we focused on the potential of methionine, which is an essential sulfur-containing amino acid [8], constituting the primary source of cysteine for the synthesis of glutathione (GSH) via methionine adenosyltransferase (MAT), S-adenosyl-L-homocysteine hydrolase (SAHH), cystathionine *β*-synthase (CBS), cystathionine *γ*-lyse (CTH), glutamate-cysteine ligase (GCL), and glutathione synthase (GS) [9]. Methionine performs an essential role in stimulating GSH synthesis by decreasing ROS accumulation and decreasing ROS levels thanks to the methionine sulfoxide reductase (Msr) system. In this regard, Liu et al. [10] reported that MsrA prevents dopaminergic neurons death and protein aggregation caused by complex I inhibition and *α*-synuclein over-expression [10].

Another amino acid that raised interest for its antioxidant potential is taurine. Taurine is a beta-amino acid with a sulfonate group and lacking carboxyl group. Diet is the main source of taurine, but it is also endogenously synthesized via different synthetic pathways starting from cysteine, homocysteine, and cysteamine [11]. In the last ten years, a growing number of reports contributed to characterize the various biological effects that taurine exerts in different organs and systems (i.e., retina, kidney, central nervous system, and cardiovascular system). The abundance of taurine in all brain regions, especially in the developmental stage, arouses a strong interest in this amino acid’s role in brain development and cognitive function. Therefore, several studies investigated taurine administration’s efficacy in attenuating nervous system disorders [12], using animal models of Alzheimer’s disease, brain ischemia [13], and Huntington’s disease [14,15,16]. Interestingly, regarding PD, a recent paper reported that taurine protected dopaminergic neurons in a mouse model of PD by inhibiting microglia-mediated neuroinflammation [17]. Even though the exact mechanisms of action are not yet fully elucidated, taurine has been clearly shown to be a key regulator of superoxide levels [18,19] with a significant cytoprotective role towards mitochondria and endoplasmic reticulum stress in several neural systems, including primary neuronal cultures and human neuroblastoma cell lines [20,21,22]. Further than its antioxidant activity, accumulating evidence indicates that taurine can directly or indirectly act on several key apoptotic pathways [19], favoring the restoration of the cellular Bcl-2 pool, preventing the release of cytochrome c from mitochondria, and suppressing the assembly of the apoptotic protease activating factor-1 (Apaf 1)/caspase-9 apoptosome complex, thus avoiding caspase-9 activation [23].

Based on this evidence, we studied the potential effects of the amino acid L-methionine on an in vitro model of PD comparing it to the known antioxidant taurine. In particular, we tried to dissect the underlying mechanisms to gain insights into the potential of methionine treatment in slowing the progression of the disease by maintaining mitochondrial functionality. In addition, to ascribe these effects to mitochondria and oxidative stress, especially for methionine, Met-O was used in place of methionine because in the presence of a strong oxidant it is further irreversibly oxidized to form methionine sulfone (Met-O_2_) [24].

## 2. Materials and Methods

### 2.1. Cellular Model

The human neuroblastoma cell line SH-SY5Y (ATCC Cat. CRL-2266) was cultured in Dulbecco’s minimum essential medium (DMEM) (Corning^®^, Corning, NY, USA) supplemented with glutamine and antibiotics. Differentiated SH-SY5Y cells are broadly accepted for in vitro studies requiring neuronal-like cells. Differentiated SH-SY5Y cells are low-cost to culture, and the ethical concerns related to primary human neuronal culture are avoided. Moreover, since SH-SY5Y cells are human-derived, they express several human-specific proteins and isoforms that would not be inherently present in rodent primary cultures [25,26]. To induce the differentiation, SH-SY5Y were plated at 15,000 cells/cm^2^ and after 24 h were grown in 10 μM Retinoic acid (RA) (Sigma, St. Louis, MO, USA) with 1% FBS for 3 days. After 3 days, the media were replaced with fresh 12-*O*-tetradecanoylphorbol-13-acetate (TPA) (Sigma, USA) for another 3 days of differentiation. For the setup of the in vitro Parkinson’s disease model, the SH-SY5Y cells line was treated with 6-OHDA (Sigma, USA) (ranging from 25 µM to 100 µM) for 24 h after differentiation [27]. The treatments with L-methionine and taurine (Dompé Farmaceutici, Milano, Italy), ranging from of 0.1 mg/mL to 5 mg/mL, were performed one hour before 6-OHDA stress.

### 2.2. Cell Viability

Cell viability was determined using Cell Titer One Solution Cell Proliferation Assay (Promega, Madison, WI, USA), as previously described [28], and measured at 492 nm using an ELISA plate reader, Infinite F200 (Tecan, Männedorf, Switzerland). Cells were treated with L-methionine or taurine both at 0.1 mg/mL for one hour and after with 6-OHDA at 35 µM concentration for 24 h. The assay was performed in quintuplicate. The results were expressed as absorbance at 492 nm.

### 2.3. Annexin V

The apoptosis assay was performed with the IncuCyte Live System (Sartorius, Gottingen, Germany); cells were seeded on a multiwell plate at a density of 1.5 × 10^4^ cells for well. After treatments, cells were incubated with the Alexa Fluor 488 Annexin V (1:200; Cat. V13241) in annexin binding buffer (Cat. V13241) for 15 min at 37 °C. After incubation, the multiwell was transferred to the IncuCyte controller (Sartorius) for 4 h. Data analysis was performed with the software IncuCyte S3 2018 and based on the green area/phase area’s fluorescence intensity.

### 2.4. Western Blotting

The cells were washed with cold 1× phosphate-buffered saline buffer followed by detachment with scrapers. Pellets were collected and resuspended in cell lysis buffer containing protease and phosphatase inhibitor. After, the lysate was collected and maintained on ice for 30 min. Finally, the total protein was extracted by centrifugation at 14,000× *g* RPM for 30 min at 4 °C. Protein concentration was assayed by the micro-BCA kit (Invitrogen, Waltam, MA, USA) according to Brandolini et al. [29].

SDS-polyacrylamide gel electrophoresis was performed, running samples (15–30 µg/lane) on 8–15% polyacrylamide denaturing gels. Proteins were transferred to polyvinylidene difluoride membranes and then blocked in skimmed milk for 1 h. The membranes were then incubated with: anti-cleaved caspase 9 (1:1000 Cell Signaling, Danvers, MA, USA); anti-p-JNK (1:500 Santa Cruz Biotechnology, Dallas, TX, USA); anti-p-AKT; anti-PI3K; anti-p-Bcl2 (1:1000 Cell Signaling, Beverly, MS, USA); rabbit monoclonal anti-p-Creb (1:1000 Cell Signaling, USA); anti-mBDNF (Abcam, Cambridge, UK, 1:500); anti-p-TrkB (Abcam, UK, 1:1000); anti-p-Nrf2 (Abcam, UK, 1:2000); anti-Mn-Sod2 (ThermoFischer, Waltam, MA, USA, 1:2000); anti-Catalase (Rockland, USA, 1:1000); anti-Gpx (Abcam, UK, 1:1000); anti-GstP1 (ThermoScientific, USA, 1:1000); anti-HNE (Abcam, UK, 1:500); anti-Sirt1 (Abcam, UK, 1:500); anti-p-Foxo3a (Abcam, UK, 1:1000); anti-Msrb2 (Santa Cruz, USA, 1:500), anti-Drp-1 (Santa Cruz, USA, 1:500), anti-Opa-1 (Santa Cruz, USA, 1:500), and anti-Mfn1/2 (Abcam, UK, 1:500) at 4 °C overnight. After three washes (5 min each) in TBS-T, they were further incubated with 1:20,000 horseradish peroxidase-conjugated anti-rabbit IgG or anti-mouse IgG. According to the manufacturer’s instructions, the protein bands were visualized by ECL Prime (ThermoScientific, USA) Western blotting detection reagent. Alliance Q9 MINI WL (Uvitec, Cambridge, UK) was used to image chemiluminescent bands. To perform the analysis of each band intensity the NIH ImageJ program was used. Anti-β-actin (HRP-conjugate, Cell Signaling, USA) (1:10,000) or GAPDH (1: 4000, Santa Cruz, USA) was used as the loading control depending on the molecular weight of the proteins assayed. The relative peak intensity of each marker band analyzed was normalized with respect to the loading control band.

### 2.5. OxyBlot Detection Kit

Protein carbonyls were analyzed by Western blot analysis according to the manufacturer’s protocols (OxyBlot Protein Oxidation Detection Kit, Sigma, USA). A total of 30 μg protein from protein lysate was mixed with 2,4 dinitrophenylhydrazine and blotted using a primary antibody specific to dinitrophenylhydrazone-derivatized residues (OxyBlot™) and a secondary antibody against the first one (OxyBlot). Protein carbonyls were visualized by Alliance Q9 MINI WL and revealed using enhanced chemiluminescence and quantified by densitometry using ImageJ software. The bands obtained were normalized by Coomassie blue staining of the total gel.

### 2.6. Mitotracker Deep Red and Quantification of Mitochondrial Fragmentation

MitoTracker is a fluorescent dye that stains mitochondria in live cells, and its accumulation is dependent upon membrane potential. In particular, cells were seeded on the coverslips at a density of 1.5 × 10^4^. After treatments, the cells were washed three times with PBS. Then, the cells were incubated with Mitotracker deep Red (Invitrogen, USA) in HBSS buffer at concentration of 1 µM for 15 min. Then, cells on coverslips were washed three times in PBS and then fixed with 3.7% formaldehyde in PBS for 15 min at RT. After further washes in PBS, coverslips were mounted with Vectashield mounting medium containing DAPI nuclear dye. The observation was performed with a confocal microscope Leica TCS SP5, and images were acquired with LCS Leica confocal software SP5 (Leica microsystem, Wetzlar, Germany). Quantitative determination of mitochondrial fragmentation was performed as previously described [30]. Mitochondria shorter than 2 μm were considered fragmented, and filamental mitochondria were those longer than 5 μm. The mitochondrial images were binarized by the threshold module using ImageJ, and these binary images were converted to images 1 pixel wide by the skeletonize module. Regarding mitochondrial length, it was calculated utilizing the analyze particles module. All the analyses were performed by an investigator blind to the experiment on at least 25 randomly chosen cells [31].

### 2.7. Mitosox

Subsequently, cells were stained with Mitosox Red reagent, which is a novel fluorogenic dye specifically targeted to mitochondria in live cells. Mitosox Red reagent permeates live cells where it selectively targets mitochondria. It is rapidly oxidized by superoxide but not by other reactive oxygen species (ROS) and reactive nitrogen species (RNS). When the red reagent is oxidized by superoxide, it exhibits red fluorescence. Cells were marked with Mitotracker Green FM, washed three times with PBS, and then incubated with MitoSox (Invitrogen, USA) in HBSS buffer at 1 µM concentration for 10 min. After different washes in PBS, coverslips were mounted with Vectashield mounting medium containing DAPI nuclear dye. The observation was performed with a confocal microscope Leica TCS SP5, and images were acquired with LCS Leica confocal software SP5 (Leica microsystem, Wetzlar, Germany). The data analyses were performed using NHI ImageJ. The fluorescence intensity for Mitotracker green was normalized on the nuclei (DAPI), while the fluorescence intensity for Mitosox Red was normalized on the normalized Mitotracker intensity.

### 2.8. TMRM

Healthy mitochondrial membranes maintain a difference in electrical potential between the interior and exterior of the organelle, referred to as a membrane potential. Tetramethylrhodamine methyl ester (TMRM) is a cell-permeant dye that accumulates in active mitochondria with intact membrane potentials. If the cells are healthy and have functioning mitochondria, the signal is bright. Upon loss of the mitochondrial membrane potential, TMRM accumulation ends, and the signal fades or disappears. The TMRM assay was performed with the IncuCyte Live System. The cells were seeded on the multiwell plate at a density of 2 × 10^4^ cells per well (optimized to have a better staining). After treatment, the cells were incubated with TMRM reagent (200 Nm, Invitrogen, USA) in PBS for 15 min at 37 °C. After incubation, the multiwell was transferred to the IncuCyte controller for 24 h.

### 2.9. Seahorse Mito Stress Assay

The Seahorse XF96e Extracellular Flux Analyzer (Agilent Technologies, Santa Clara, CA, USA) was used to generate the bioenergetic profiles of differentiated neuroblastoma SH-SY5Y cell line upon treatments. Live-cell analyses of oxygen consumption rate (OCR) and extracellular acidification rate (ECAR) were measured using the Mito Stress test (Agilent). Cells were cultured on a Seahorse XF96 cell culture plate (Agilent, USA) at a density of 5 × 10^4^ cells/well (cell density was optimized to ensure a proportional response of FCCP with cell number) and grown overnight in DMEM 10% of FBS, then differentiated as described above. After complete differentiation, cells were treated as described above. On the day before the Seahorse assay, the cartridge was hydrated and incubated overnight at 37 °C in the absence of CO_2_. On the day of the assay, cell medium was replaced with unbuffered DMEM (XF Assay Medium; Agilent Technologies, USA) supplemented with 5 mM glucose and 1 mM sodium pyruvate (Agilent Technologies, USA), and incubated for 1 h at 37 °C in the absence of CO_2_. Medium and reagents were adjusted to pH 7.4 on the day of the assay. After four baseline measurements for the oxygen consumption ratio, cells were sequentially challenged with injections of Mito Stress drugs prepared following the manufacturer’s instructions. The final concentrations used for each drug were 1 μM oligomycin (ATP synthase inhibitor), 1 μM FCCP (mitochondrial respiration uncoupler), and 0.5 μM rotenone/antimycin (complex I and III inhibitors). For the normalization in port D, Hoechst 33,342 solution was injected, and at the end of the run, the plate was read using a microplate reader (Infinite Tecan, Männedorf, Switzerland). The data and graphs generated at the end of the Mito Stress assay were extracted using Wave software.

### 2.10. SOD Activity Assay Kit

The total SOD activity was measured by an enzymatic method using an SOD assay kit purchased from Abcam, UK. SOD activity was assessed by measuring the rate of reduction of WST-1 (2-[4-lodophenyl]-3-[4-nitrophenyl]-5-[2,4-dis-ulfophenyl]-2H-tetrazolium, monosodium salt), which produces a water-soluble formazan dye upon reduction with a superoxide anion. The absorbance of WST-1 was measured at 450 nm. The total measured SOD activity was calculated using the decrease in color development at 450 nm compared to the control.

### 2.11. Expression Analysis: RNA-Seq and Gene Clustering

Cells were cultured as described, and RNA was extracted with Trizol (Invitrogen, USA) following the manufacturer’s instructions. Data were obtained from SH-SY5Y untreated cells as a control and cell treated either with 6-OHDA alone or in combination with Met, Met-O, or Tau. All the experiments were performed in triplicate. RNA-seq experiments were performed at Lexogen, a biotech company expert in expression profiling technologies using the QuantSeq 3′ mRNA-Seq Library Prep Kit, which is a library preparation protocol designed to generate libraries of sequences close to the 3′ end of polyadenylated RNA. Data were aligned by the STAR tool (https://github.com/alexdobin/STAR/blob/master/doc/STARmanual.pdf, accessed on 5 June 2021), and differential expression analysis was performed. DESeq2 is a bioconducter package implemented in R (http://www.bioconductor.org/packages/release/bioc/vignettes/DESeq2/inst/doc/DESeq2.html#theory, accessed on 5 June 2021). Aggregated quality control and trimming to remove various types of unwanted sequences (i.e., primers, poly-A tails) were performed with MULTIQC and Cutadapt tools, respectively (http://multiqc.info/, accessed on 5 June 2021; https://cutadapt.readthedocs.io/en/stable/, accessed on 5 June 2021). Clustering and functional analysis of differential expression genes was performed by CHIMERA (Metrica e cluster construction). Interactors’ network was built with STRING (http://string-db.org, accessed on 5 June 2021).

### 2.12. Bioinformatic Analysis

Database for Annotation, Visualization and Integrated Discovery (DAVID) v6.8 was used to perform Gene Ontology (GO) enrichment analysis [32]. We performed a functional annotation clustering for GOTERM_BP 4 and 5 and selected the GO terms with an FDR < 0.05.

STRING is a database of known and predicted protein–protein interaction (PPIs), in which interactions are derived from five main sources: genomic context predictions, high-throughput lab experiments, (conserved) co-expression, automated text mining, and previous knowledge in databases [33]. For this analysis, we used the following parameters:-Network edges represent data from high-throughput lab experiments, (conserved) co-expression, and previous knowledge in databases.-The interaction score cutoff used was 0.7 (high confidence).-The list of proteins used was generated starting from the expression data achieved by RNA-seq. The log(2) of the 6-OHDA+drug (methionine or taurine) compared to 6-OHDA alone was calculated as the difference between log(2) observed in the 6-OHDA+drug and 6-OHDA samples compared to untreated cells, respectively. For the analysis we used the genes with a log(2) >of 0.4 and <−0.4; these genes are mentioned in the text as “recovered”.

### 2.13. Statistical Analysis

Prior to the analysis, the Shapiro–Wilk test was used to assess the normal distribution of the samples. Data are mean ± SE of 3 different biological experiments. Statistical analysis was performed by one-way ANOVA following Tukey’s post hoc test. For grouped analyses (TMRM), two-way ANOVA was used followed by Tukey’s post hoc test. The level of significance was set at *p* < 0.05.

## 3. Results

### 3.1. Methionine and Taurine Counteract 6-OHDA Injury

In a preliminary series of experiments, cell viability dose–response curves were performed for 6-OHDA, Met, taurine, and Met-O, to determine the best concentration to be used in subsequent experiments (Appendix A). On this basis, 35 µM for 6-OHDA and 0.1 mg/mL for Met, taurine, and Met-O have been used in the following experiments.

As illustrated in Figure 1, cell viability was significantly increased in the presence of 0.1 mg/mL of Met or taurine compared to the stressed conditions. As described in the introduction, Met-O was used in place of Met since is generally not able to act as an antioxidant.

### 3.2. RNA-Sequencing Analysis

To shed light on the protective effects of Met and taurine in counteracting 6-OHDA-induced injury, we decided to perform RNA-sequencing experiments on differentiated SH-SY5Y cells treated with 6-OHDA, a broadly used cellular model for PD. The analysis showed that more than 3000 transcripts were dysregulated by 6-OHDA (adjusted *p* < 0.05). Interestingly, cluster analysis revealed that the general pattern of gene regulation afforded by 6-OHDA is reverted by Met and taurine addition, while Met-O, as expected, was not able to recover the general pattern of gene expression dysregulated by 6-OHDA (Figure 2). In particular, the MA plot shows that taurine and Met led to significant changes in gene expression compared to control (CTR), with a lower number of genes with respect to 6-OHDA and Met-O (Figure 2).

Gene expression relationships provide important clues about gene function; thus, we decided to explore the behavior of groups of genes involved in mitochondrial functions, apoptosis, Parkinson’s disease, and oxidative stress, which represent relevant mechanisms in our experimental model.

Of the 99 genes involved in the morphological and physiological alterations undergone by mitochondria during apoptosis, about one-third were significantly changed by Met and taurine treatments (Figure 3, GOBP_Apoptotic_Mitochondrial_Changes). Interestingly, among this class of genes, SOD2, which resulted downregulated by 6-OHDA, was clearly recovered by Met treatment. Other groups of about 100 transcripts dysregulated by 6-OHDA were involved in mitochondrial Cytochrome C oxidase assembly, regulation of mitochondrial membrane potential, and permeabilization (Figure 3, GOBP_Mitochondrial_Cytochrome_C_Oxidase_ Assembly and GOBP_Regulation_ Mitochondrial_Membrane_Potential). Thirty of these transcripts were significantly modulated by 6-OHDA and reverted by Met treatment; moreover, some downregulated transcripts such as Cox17 and Bcl2 were partially recovered by Met treatment and in a lower manner by taurine treatment, suggesting that Met could be more effective in addressing some specific gene groups. Taken together, our results suggested that Met can recover the transcription of genes involved in oxidative imbalance and mitochondrial dysfunction observed in 6-OHDA treated cells.

As a proof of principle, we also looked specifically at the expression of PD-associated transcripts and FoxO-mediated transcription of oxidative stress, metabolic and neuronal genes, since FoxO is one of the key regulators of neurodegeneration [34]. In the group of PD transcripts, the pattern of expression of almost all the genes identified was recovered by both Met and taurine with a strong effect on a number of transcripts such as Cox5B and Slc25A6 (Figure 3, KEGG_Parkinsons_Disease). The same general pattern was observed for FoxO-associated transcripts and in particular for some Smad proteins and Foxo3 (Figure 3, Reactome_FOXO).

Finally, since Msr enzymes are key regulators of oxidative stress in the cells, we decided to build a protein–protein interaction network by STRING (http://string-db.org, accessed on 5 June 2021) around the Msr proteins (MSRA, MSRB2 and MSRB3) (Figure 3, STRING Mrs Protein) and looked at the expression of the respective transcripts observed in the protein network. Interestingly, neither MSRA nor MSRB had differential expression after 6-OHDA treatment alone and in combination with the other drugs, while MSRB2 was downregulated by 6-OHDA and recovered by the treatments (Appendix A). Together with MSRB2, other interactors were clearly recovered by Met and taurine, confirming the role of this pathway in Met-associated regulation of oxidative stress.

We, then, decided to explore by bioinformatics analysis the functional enrichment of the list of genes dysregulated by 6-OHDA treatment and recovered by Met or taurine. The analysis was performed by STRING database. STRING generates PPI networks, confirming the enrichment for GO terms and protein interactions associated with apoptosis/cell death and neurogenesis mainly among the transcripts respectively up-or down-regulated by 6-OHDA and recovered by the treatments, as reported in Appendix A. Interestingly, among the genes recovered by Met and taurine, there are members of the sirtuins and neurotrophins family, such as Sirtuin 2 (SIRT2) and brain derived neurotrophic factor (BDNF), as well as Bcl2 family members, which are crucial in neurodegenerative processes [35,36]. Methionine can reduce ROS levels through the activity of the Msr system, in which methionine sulfoxide reductases (e.g., MsrA, MsrB) act as natural scavenging systems for ROS by catalyzing the conversion of Met-O to Met. Met-O is generally not able to work as an antioxidant.

All together, these results suggested that Met could exert protective effects in an in vitro model of PD, possibly protecting neurons. We then performed further experiments to validate this hypothesis.

### 3.3. Methionine and Taurine Counteract 6-OHDA-Induced Apoptosis

Apoptosis evaluation by Annexin V live assay by IncuCyte system and Western blotting (WB) analysis for cleaved-caspase 9 and p-JNK are reported in Figure 4. 6-OHDA strongly increased Annexin staining as well as cleaved-caspase 9 and p-JNK protein levels; treatment with Met or taurine significantly decreased the apoptotic markers, while Met-O was ineffective in counteracting the more marked Annexin staining induced by 6-OHDA.

### 3.4. Methionine and Taurine Effects on the Survival and Neuroprotective Pathways

The survival pathway (p-AKT, PI3K, p-Bcl2) and the neuroprotective pathway (p-CREB, mBDNF, p-TrkB) assayed in control and treated cells are shown in Figure 5. 6-OHDA sharply decreased survival proteins and neurotrophin signaling. In contrast, Met or taurine almost restored the control levels of the analyzed proteins. The treatment with Met-O after 6-OHDA did not restore the protein levels for p-CREB and was unable to activate the survival pathway PI3K/AKT.

### 3.5. Methionine and Taurine Effects on the 6-OHDA-Induced Oxidative Stress Signaling Pathways

In Figure 6, the antioxidant Nrf2 signaling and lipid peroxidation products (4-HNE) were analyzed by WB, while oxidized proteins were assayed by Oxyblot detection kit, and SOD activity was measured by an enzymatic method. 6-OHDA markedly decreased Nrf2 signaling while increasing oxidized proteins and lipid peroxidation products. Met and taurine counteracted these effects, while Met-O was unable to induce antioxidant enzymes, such as catalase, Mn-Sod2, Gst-P1, and Gpx1, by p-Nrf2 and to counteract the toxic effects of ROS. These data were also confirmed by SOD enzyme activity.

In Figure 7, the MsrB2 pathway is shown. Sirt1, p-Foxo3a, and MsrB2 were analyzed by WB analysis. Once activated, Sirt1 triggers the nuclear localization of FOXO3a, which in turn increases MsrB2 transcription [37]. In our experimental conditions, 6-OHDA significantly decreased this pathway, while Met was the only compound able to counteract this effect. Indeed, Met-O and taurine did not increase Sirt1, p-FOXO3a, and MsrB2 levels and transcripts.

### 3.6. Methionine and Taurine Preserve Mitochondrial Functionality and Morphology

Mitochondrial functionality was assayed in all the conditions tested (Figure 8, Figure 9 and Figure 10). In particular, 6-OHDA alone or in combination with Met-O showed significant increases in MitoSox red fluorescent emission, suggesting an increase in superoxide in mitochondria, while Met and taurine were able to reduce red fluorescent emissions. These results suggest that Met and taurine decreased 6-OHDA-induced superoxide generation in mitochondria (Figure 8).

TMRM was used to test the integrity of mitochondrial membrane potential, which is accumulated in active mitochondria (Figure 9). 6-OHDA significantly decreased TMRM fluorescence intensity (meaning abnormal membrane potential), while Met and taurine were able to maintain mitochondrial membrane potential. Interestingly, TMRM fluorescence intensity started to decrease at 4h after 6-OHDA treatment, and both Met and taurine showed protective effects in maintaining normal mitochondrial membrane potential as early as 4 h after the 6-OHDA insult (Figure 9).

Healthy mitochondrial morphology and dynamics are essential for maintaining mitochondrial function; for this reason, Mitotracker deep Red was used to evaluate mitochondrial fragmentation. The mitochondrial morphology of our cell model (without any treatment, CTR) showed tubular networks, while upon 6-OHDA challenge, the mitochondria were mainly shorter and smaller size, suggesting mitochondrial fragmentation. Notably, mitochondrial morphological changes were attenuated by Met and taurine treatments (Figure 10).

Mitochondrial morphology is likely to be the result of the interplay between mitochondrial division and fusion. Mitochondrial fusion is controlled by Mfn1/2 and Opa-1, whereas mitochondrial fission is mainly controlled by Drp-1. To study whether mitochondrial fission and fusion were influenced by 6-OHDA and the tested compounds, the expression of these factors was assayed by Western blot (Figure 10). We found a significant reduction in Opa-1 and Mfn1/2 upon 6-OHDA stress, and Met and taurine prevented the decrease in these fusion proteins. As for Drp-1, which is a fission protein, the protein levels were increased by 6-OHDA, while both Met and taurine were able to partially counteract this effect (Figure 10).

In line with previous results, Met-O presence was ineffective in producing the protecting effects observed for Met on mitochondrial functionality (Figure 8, Figure 9 and Figure 10).

### 3.7. Methionine and Taurine Rescue 6-OHDA-Altered Bioenergetic Profile

We then investigated the mitochondrial functionality in our cell model using Mito Stress assay by Seahorse Extracellular Flux Analyzer to characterize the cells’ bioenergetics profile upon treatments (Figure 11). In particular, in Figure 11 the time course and live measurement of OCR with injections of stressor compounds are shown. OCR resulted strongly reduced by 6-OHDA, whereas the co-presence of Met and taurine was able to counteract this effect in contrast with Met-O (Figure 11A).

With the injection of oligomycin, cells treated with 6-OHDA and Met as well as taurine maintained much higher oxygen consumption than 6-OHDA alone or Met-O, leading to significant differences in ATP production through mitochondrial respiration and proton leak (Figure 11B). Induction of maximal respiration with FCCP injection resulted in decreased oxygen consumption upon 6-OHDA. The combined treatment showed a behavior similar to the CTR condition, while Met-O was similar to cells upon 6-OHDA. Furthermore, 6-OHDA-treated cells had considerably reduced non-mitochondrial respiration, while cells upon Met and taurine in combination with 6-OHDA showed the same behavior as control cells in contrast with Met-O. With the injection of rotenone/antimycin A, cells upon 6-OHDA showed an extremely low spare respiratory capacity, while the presence of methionine and taurine significantly counteracted this effect (Figure 11B).

## 4. Discussion and Conclusions

Dopaminergic neurons are already damaged at the time of diagnosis of PD and the first appearance of symptoms. Therefore, the interest in protective treatments aimed at decreasing oxidative stress and delaying disease progression is very high. In this regard, it is worth considering that oxidative stress, mitochondrial dysfunction, and redox balance perturbation are observed in neurodegenerative diseases, including PD [38,39,40]. Refs. [41,42,43,44,45] Methionine has a pivotal role in GSH synthesis, decreasing ROS accumulation. Methionine can also reduce ROS by the Msr system (MsrA, MsrB), which works as a scavenger by catalyzing the conversion of Met-O to methionine [46] (Appendix A).

In addition, Wang et al., 2019 [47], demonstrated that oral supplementation of methionine increased antioxidant enzymes such as manganese superoxide dismutase, catalase, and glutathione peroxidase. It has also been suggested that the activation of the Nrf2-ARE pathway is dependent on the availability of L-methionine [47]. As a sulfur-containing amino acid, L-methionine is one of the most susceptible to oxidation by ROS. However, as opposed to other antioxidant molecules, the oxidation of methionine is reversible. Met-O can be reduced back to methionine by Msr and, during each cycle of oxidation and reduction, one equivalent of ROS is consumed, thus acting as an endogenous scavenging system [48]. Msr constitutes a group of ubiquitous enzymes that catalyze the reduction of free and protein-derived Met-O to Met. In terms of cellular localization, MsrA is present in the cytoplasm, nucleus, and mitochondria, while MsrB2 is present only in mitochondria [48,49]. The Msr system has been extensively studied for its antioxidant role and the protection against oxidative stress and apoptosis. ROS can directly oxidize amino acids, and methionine is readily oxidized by the Msr system to methionine sulfoxide, which can be reversed. Oxidation of methionine results in an S or R sulfoxide diastereomer [50]. The S and R methionine sulfoxides in proteins are reduced by MsrA and MsrB, respectively. In humans, MsrA has nuclear/cytoplasmic and mitochondrial isoforms generated from the MSRA gene. There are three distinct human MSRB genes: MSRB1/SELR/SELX, MSRB2/SEPX1/CBS-1, and MSRB3. MsrB1 is a selenoprotein that is primarily localized into the nuclear and cytoplasmic cellular fractions, where MsrB2 and MsrB3 are associated with mitochondria [51].

In agreement we demonstrate herein that the Msr system, in particular mitochondrial MrsB2, plays a pivotal role in cell rescue and mitochondria viability upon 6-OHDA in the presence of methionine, thus suggesting that 6-OHDA determined, in this cellular model, an increase in R sulfoxide diastereomer [50], since no known human enzyme for reduction of unbound methionine-*R*-sulfoxide has been so far detected [42]. RNAseq analysis showed that different pathways were modulated by 6-OHDA and that Met and taurine were able to counteract many of these effects. In addition, the FOXO pathway appeared affected by 6-OHDA and was restored by Met or taurine as well as the MrsB2 transcript. In agreement with the bioinformatic analysis, we provide evidence that Met is as potent as taurine in protecting dopaminergic neurons against PD-related cellular stress by preventing the activation of apoptotic pathways (Annexin, Caspase-9, p-JNK) and inducing neuroprotective (p-CREB, mBDNF, p-TrkB) and survival proteins (p-AKT, PI3K, p-Bcl2). In line with the proposed antioxidant mechanisms, both Met and taurine showed the ability to modulate Nrf2 signaling and antioxidant enzymes while decreasing oxidized proteins under the 6-OHDA insult. As compared to taurine, Met showed the peculiar and novel property to preserve mitochondrial functionality through the specific activation of the Sirt1-FOXO3a-MsrB2 cascade [52], which is pivotal in PD pathogenesis. Furthermore, Met or taurine inhibited 6-OHDA-induced oxidative stress, reducing mitochondrial fragmentation through the induction of mitochondrial fusion proteins (Mfn1/2 and Opa-1) and the inhibition of fission protein (Drp-1). In agreement with these results, Met behavior on the bioenergetic profile resulted comparable to taurine behavior in counteracting the reduction of OCR induced by 6-OHDA, as reported in Seahorse assay.

Based on these results, it seems conceivable that an L-methionine-enriched diet could show therapeutic potential for human aging and age-related disorders, protecting neurons from oxidative imbalance and maintaining mitochondrial functionality. In Figure 12, a schematic representation of the obtained results on the methionine effects is reported.

## Figures and Tables

**Figure 1 antioxidants-10-01467-f001:**
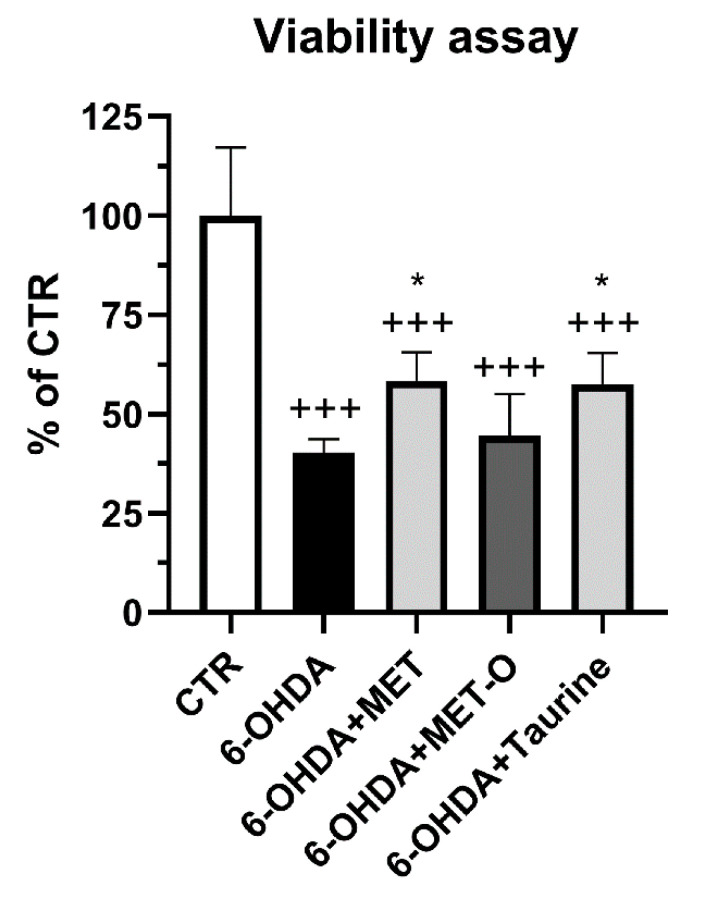
Cell viability assay for differentiated SH-SY5Y pretreated with Met or Met-O and taurine before 6-OHDA treatment. Data are mean ± SD of 3 different experiments. * *p* < 0.05 vs. 6-OHDA; +++ *p* < 0.0001 vs. control (CTR).

**Figure 2 antioxidants-10-01467-f002:**
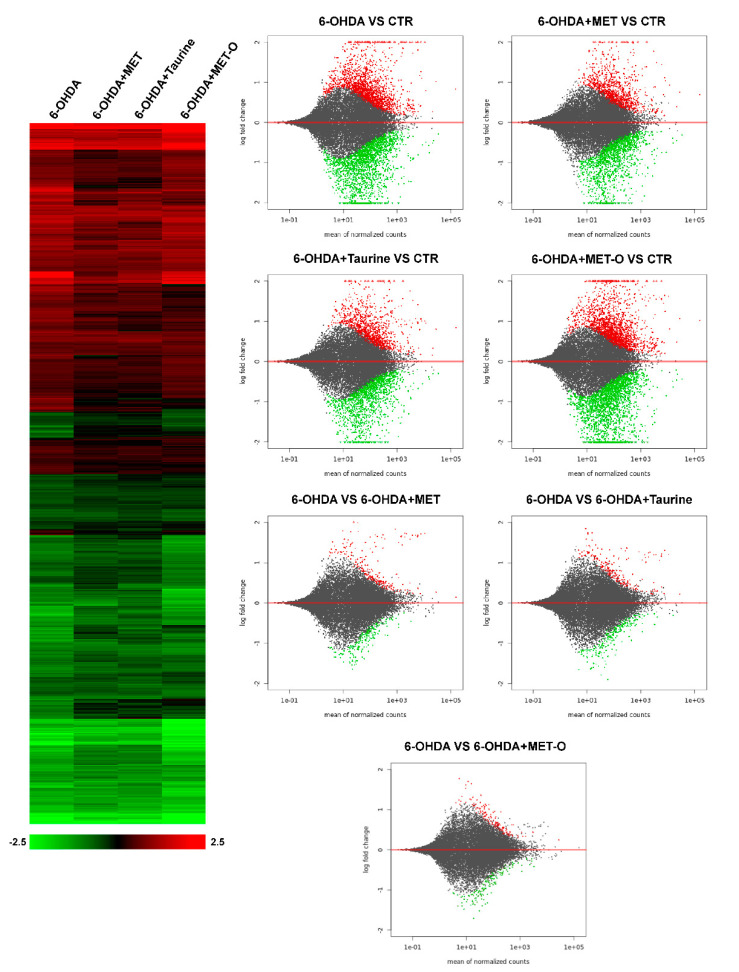
On the left, heatmap of hierarchical clustering (Euclidean metrics) of the 3175 genes whose expression is significantly perturbed by 6-OHDA (adjusted *p* < 0.05). Color scale represents log2 ratios of the expression levels in the indicated condition versus control. Color scale limits are indicated in the box below the heatmap. On the right, MA plots show up- and down-regulated genes in the different treatments (in red, the significantly upregulated genes; while in green, the downregulated).

**Figure 3 antioxidants-10-01467-f003:**
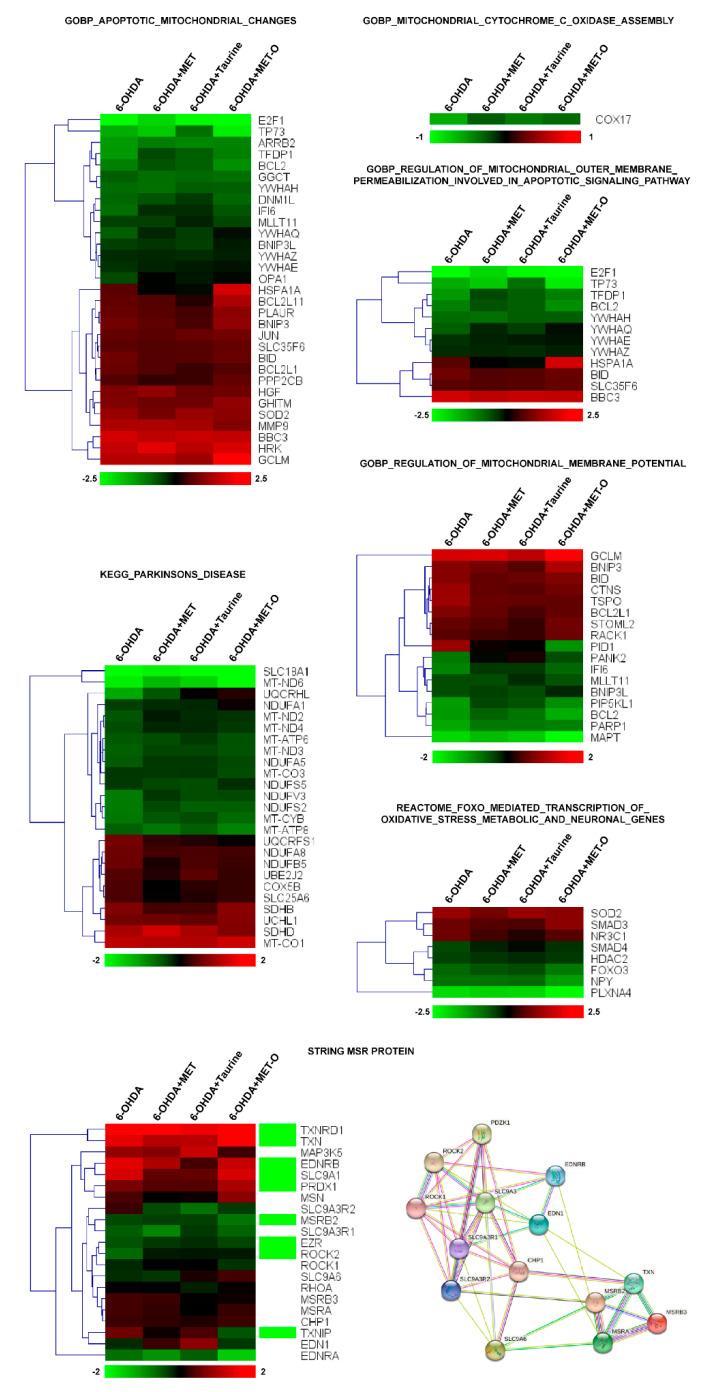
Heatmap of hierarchical clustering of the selected pathways. All the genes of the pathway that are significantly affected by 6-OHDA (adjusted *p* < 0.05) are reported in the heatmaps. Green boxes indicate the genes significantly affected by 6-OHDA. Color scale represents log2 ratios of the expression levels in the indicated condition versus control. Color scale limits are indicated in the lower box.

**Figure 4 antioxidants-10-01467-f004:**
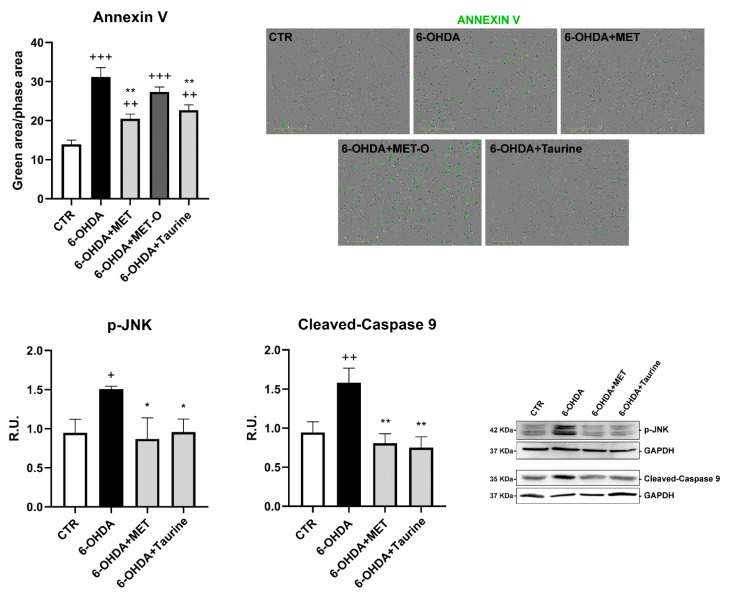
Apoptosis assayed by Annexin V in live-cell by Incucyte system and WB analysis for cleaved-caspase9 and p-JNK. A representative WB figure is shown. Data are mean ± SD of 3 different experiments. * *p <* 0.05, ** *p <* 0.005 vs. 6-OHDA; +++ *p* < 0.0001; ++ *p <* 0.005; + *p <* 0.05 vs. CTR.

**Figure 5 antioxidants-10-01467-f005:**
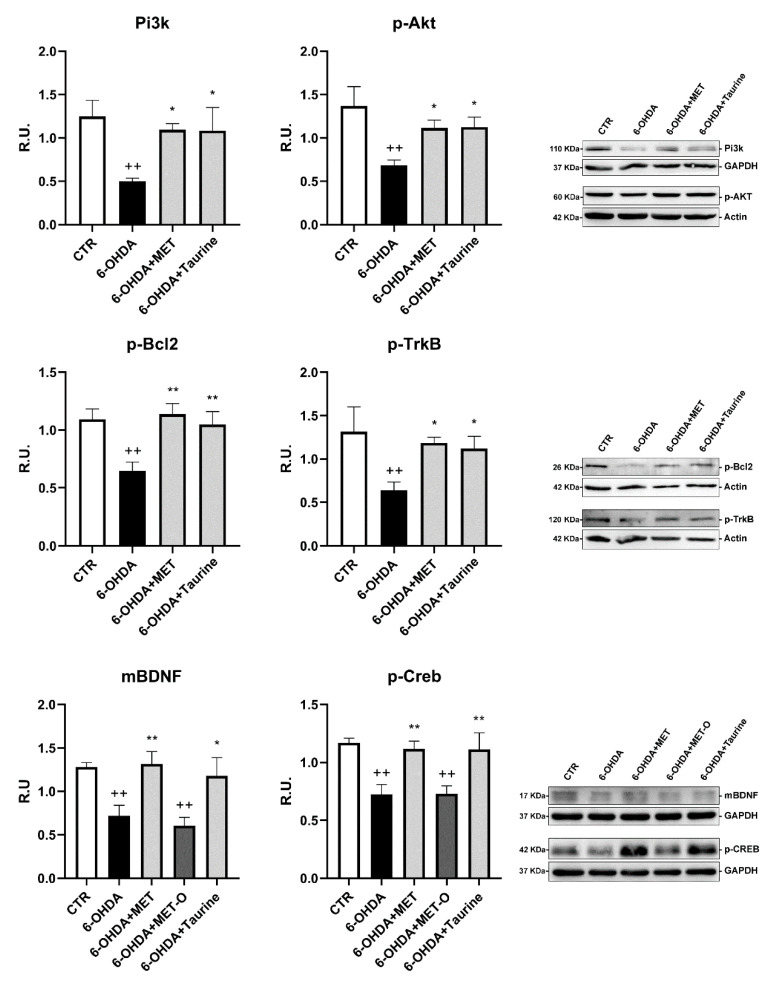
WB analyses for neuroprotective pathways. A representative WB figure is shown. Data are mean ± SD of 3 different experiments. * *p <* 0.05, ** *p <* 0.005 vs. 6-OHDA; ++ *p <* 0.005 vs. CTR.

**Figure 6 antioxidants-10-01467-f006:**
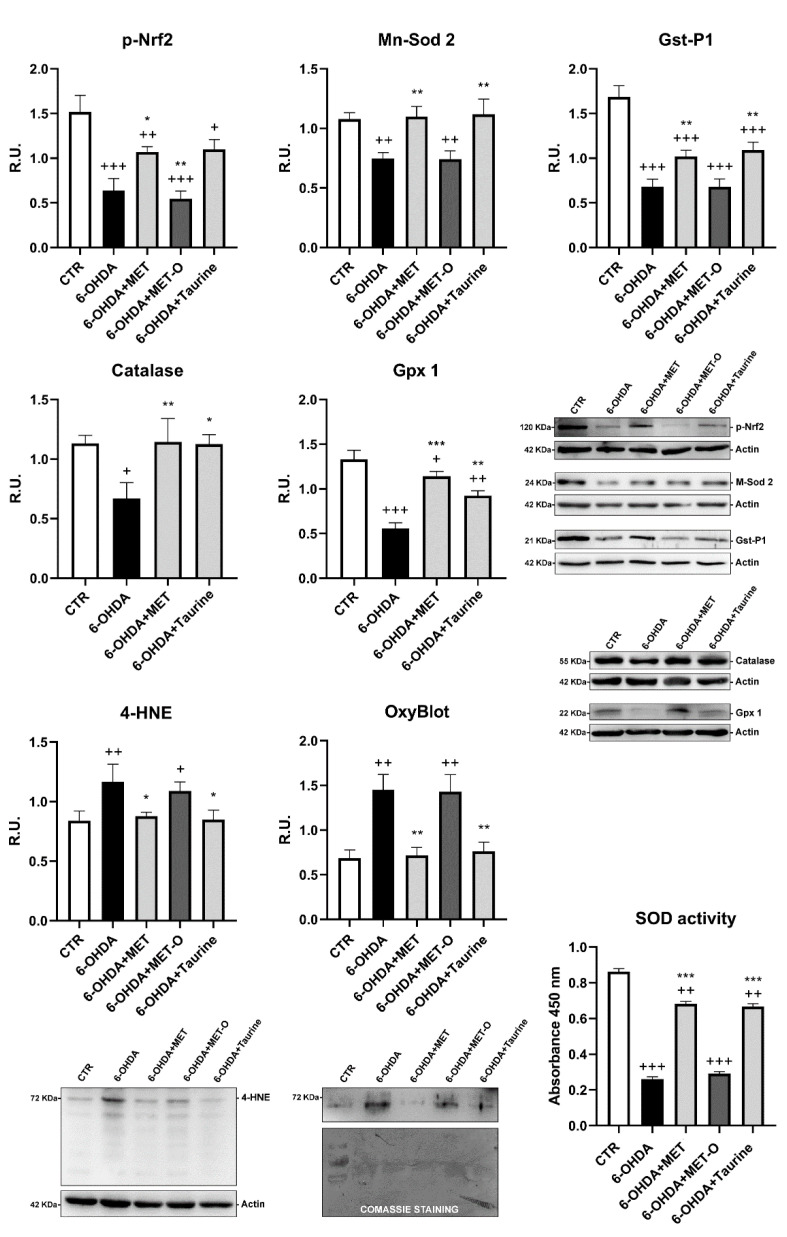
Antioxidant pathway analyzed by WB analyses, Oxyblot detection kit, and SOD activity assay. A representative WB figure is shown. Data are mean ± SD of 3 different experiments. *** *p <* 0.0001, ** *p <* 0.005, * *p <* 0.05 vs. 6-OHDA; +++ *p <* 0.0001; ++ p < 0.005; + *p <* 0.05 vs. CTR.

**Figure 7 antioxidants-10-01467-f007:**
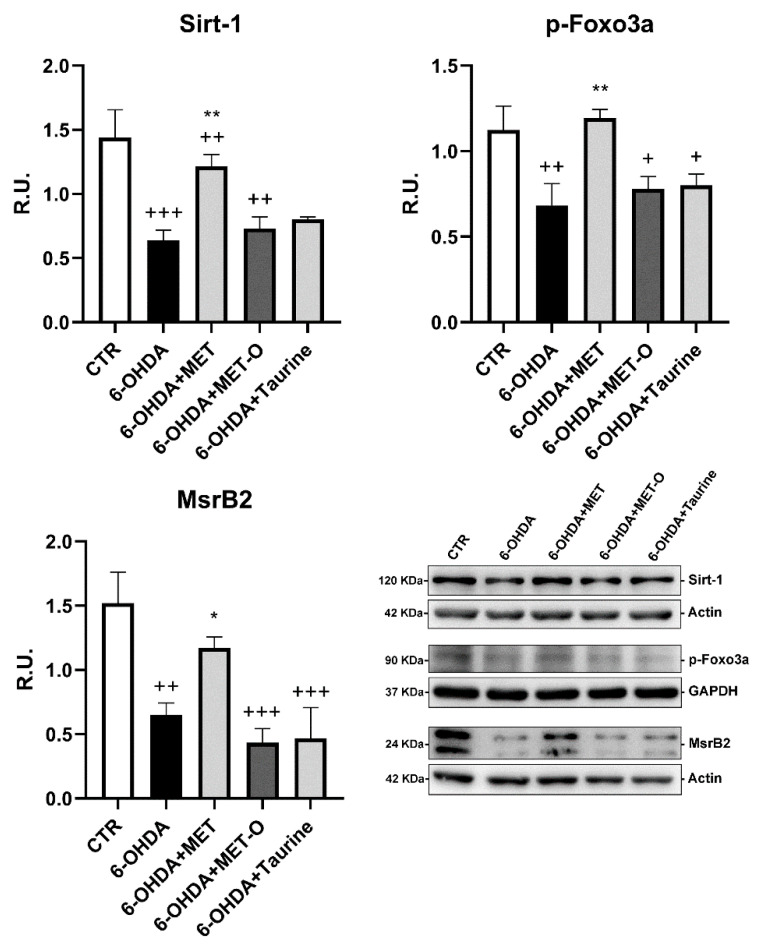
WB analyses for the Mrsb2 pathway. A representative WB figure is shown. Data are mean ± SD of 3 different experiments. ** *p <* 0.005, * *p <* 0.05 vs. 6-OHDA; +++ *p <* 0.0001; ++ *p <* 0.005; + *p <* 0.05 vs. CTR.

**Figure 8 antioxidants-10-01467-f008:**
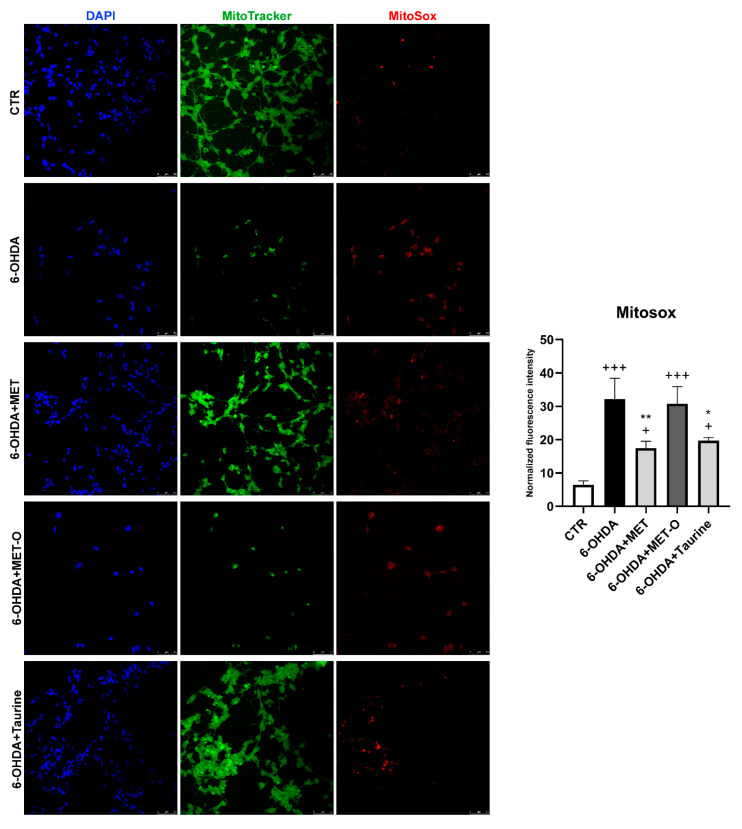
On the left, Mitotracker Green, Mitosox Red, and DAPI representative figures are shown; on the right, the normalized fluorescence intensity graph is shown. Scale bar: 50 µm. Data are mean ± SD of 3 different experiments. ** *p <* 0.005, * *p <* 0.05 vs. 6-OHDA; +++ *p <* 0.0001, + *p <* 0.05 vs. CTR.

**Figure 9 antioxidants-10-01467-f009:**
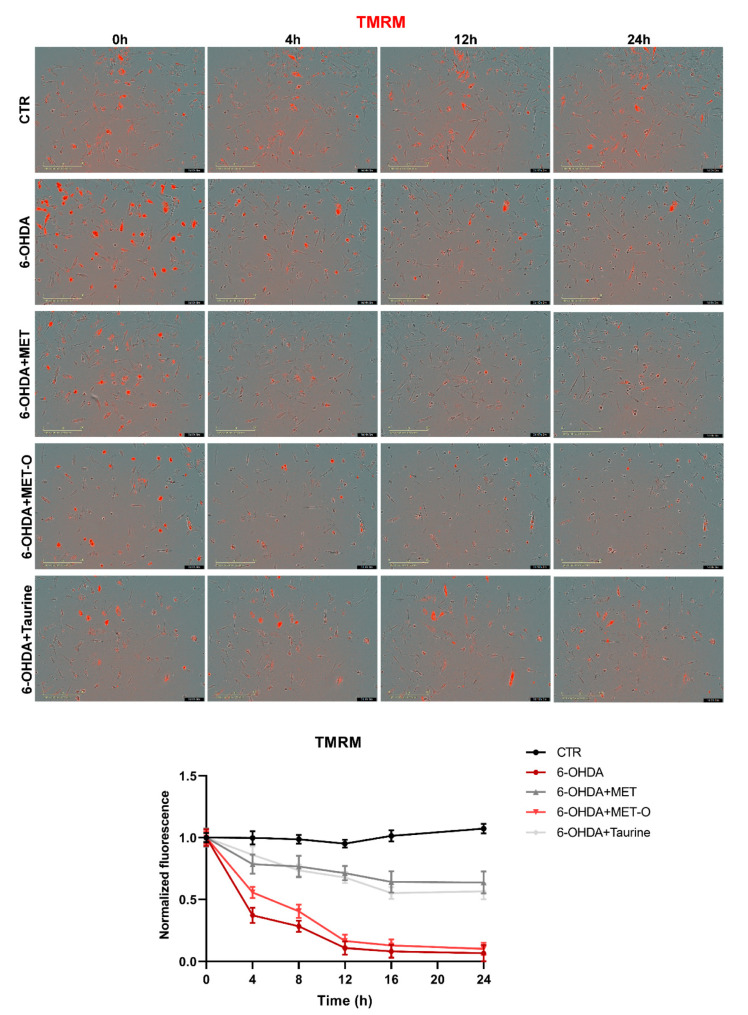
TMRM in live cells by the Incucyte system at 0, 4, 12, and 24 h time points. Representative images and relative normalized fluorescence graph are shown. Data are mean of 3 different experiments ± SD, and the significance is reported in Appendix A.

**Figure 10 antioxidants-10-01467-f010:**
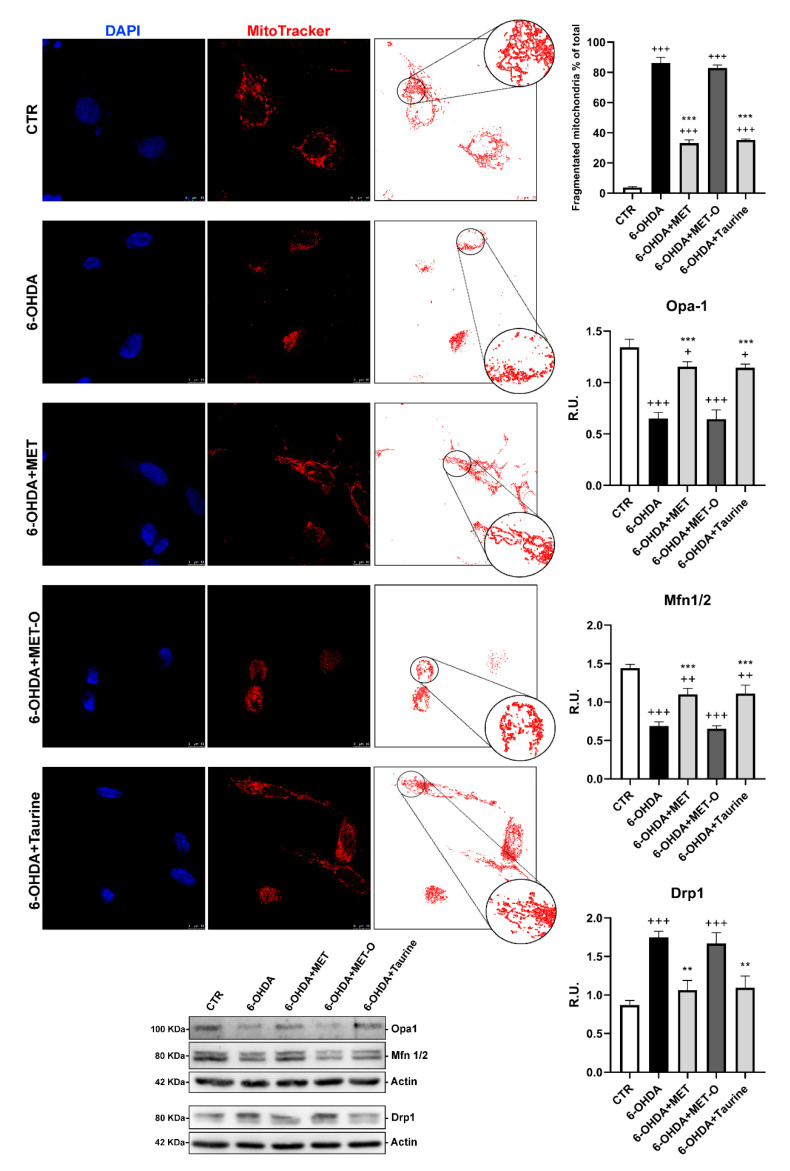
DAPI, Mitotracker deep red, and the masks generated by ImageJ representative images are reported. In the red circle the area is magnified to better appreciate the mitochondrial morphology. WB analyses for fission and fusion pathways. A representative WB figure is shown. Data are mean ± SD of 3 different experiments. *** *p <* 0.0001, ** *p <* 0.005 vs. 6-OHDA; +++ *p <* 0.0001; ++ *p <* 0.005; + *p <* 0.05 vs. CTR.

**Figure 11 antioxidants-10-01467-f011:**
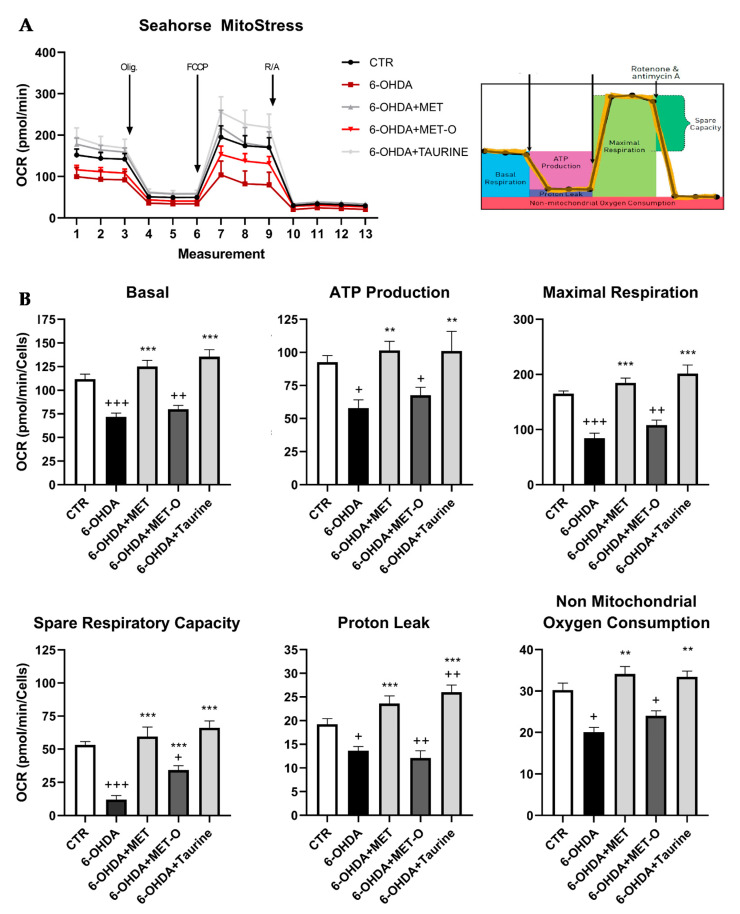
Mitochondrial bioenergetic profile in CTR and treated cells. (**A**) Seahorse XF Cell Mito stress test profile illustrated the key parameters of mitochondrial function upon the injection of the different drugs. (**B**) Graph relative to basal respiration, ATP production, maximal respiration, and non-mitochondrial respiration in control and treated cells. Data are mean ± SD of 3 different experiments. *** *p <* 0.0001, ** *p <* 0.005 vs. 6-OHDA; +++ *p <* 0.0001; ++ *p* < 0.005; + *p <* 0.05 vs. CTR.

**Figure 12 antioxidants-10-01467-f012:**
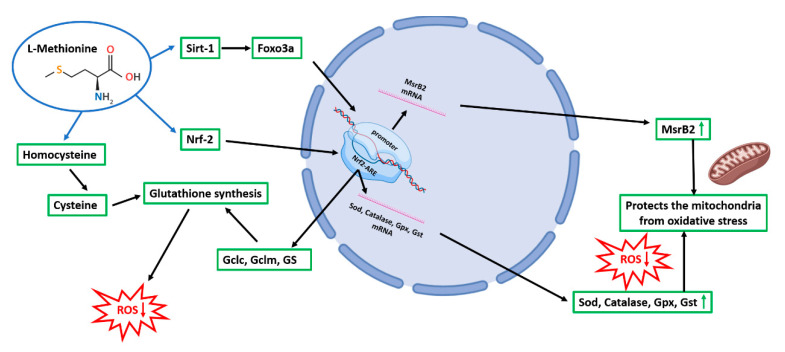
Schematic representation of the underlying mechanisms of Met neuroprotective effects. Gclc: glutamate cysteine ligase catalytic subunit; Gclm: glutamate cysteine ligase regulatory subunit; GS: glutathione synthetase.

## Data Availability

The data presented in this study are available in article and Appendix A.

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
