# Peer review of "L-Methionine Protects against Oxidative Stress and Mitochondrial Dysfunction in an In Vitro Model of Parkinson’s Disease"

_antioxidants, 2021, doi:10.3390/antiox10091467_

Round 1

Reviewer 1 Report

In this article , Catanesi et al analyzed the protective effects of the  L-methionine in an in vitro model of Parkinson’s disease and dissected the underlying mechanisms compared to taurine to gain insights into the potential of methionine treatment by maintaining mitochondrial functionality and thus slowing the progression of the disease .  Methionine sulfoxide was also used  to ascribe the effects of methionine on mitochondria and oxidative stress,. The data obtained suggested that L-methionine enriched diet could be beneficial during aging to protect neurons from oxidative imbalance and mitochondrial dysfunction thus preventing the progression of neurodegenerative processes.

The study is interesting, correctly performed and analyzed. The manuscript presents clearly the experimental procedure and data presentation is good. 

One thing that bothered me was the use of the abbreviation Tau for taurine in the text and in the figures. For me, Tau is the protein involved in Alzheimer's disease. It would be better to use another abbreviation or no abbreviation at all or to specify in the text and in the figure legends: taurine (Tau). 

line 122 : I will put the 6OHDA concentration ranging from 25µM to 100µM

line 123-124: the same for treatment with Met, taurine and Met -O (0.1mg/ml to 5mg/ml)

figure 2 : I would have liked to see 6OHDA versus 6OHDA+Met-O

line 308: something missing after "during apopotosis and"

Figure S2 legend: what is represented by green boxes is missing. Please add green boxes indicate the genes significantly affected by 6OHDA

For this figure S2 I don't understand how this figure shows that MSRA and MSRB are not affected by treatment while MSRB2 is recovered after treatment. Please clarify

Figure 4: for annexin V images, there are bars on the image. Put on the legend what bars represent

Figure 4 WB: I would like to see the effect of Met-O on p-JNK, cleaved caspase 9, Figure 5: the same for pi3K, pAKT, pBCl2, pTrkB

Figure 6: idem for catalase and Gpx1

Reviewer 2 Report

This is an interesting study in the field of neurodegeneration and neuroprotection. The antioxidant effect of L-methionine is investigated in detail in a cellular model and compared with that of taurine. A huge amount of data is presented to dissect this effect on mitochondria and oxidative stress.  However, major and minor points need to be addressed before acceptance for publication, as follows:

  1. The introduction is not well-written and the flow is quite confused. My suggestion is to organize the introduction in paragraphs that clearly focus on the main issues that the manuscript aims to deal. Although the authors place the study in a broad context, it is difficult to understand what is important. In addition, the current state of the research field is reviewed but sometimes the references are lacking or not appropriate. As an example: “….Despite decades of studies, the identification of these factors and their interaction, and the molecular mechanism of neurodegeneration that they initiate are still unclear [1].” This reference deals with the prevalence of PD, not with molecular mechanisms and open questions. Finally, the purpose of the work and its significance, including specific hypotheses being tested, are not clearly stated.

  1. Statistical analysis: did the authors perform parametric analyses (as ANOVA test is) following normality test? If yes, please indicate how the normality has been verified; if not, please analyse the data using non-parametric tests.

  1. Methods: staining with Mitotracker green and MitoSox red raises some problems. These dyes are used for staining live cells and are not well-retained after aldehyde fixation. Here, the stained cells are fixed with 3.7% formaldehyde in PBS and, after fixation, acquired and analyzed. How can the authors be sure that their analyses are reliable?

  1. Immunoblot were performed using a panel of antibodies. Why did the authors decide to use different gel loading controls, actin or alternatively GAPDH, for different blots?

  1. A crucial point is the detection of phosphorylated form of many antigens (p-JNK, p-Akt, p-BCL2, p-TrkB, p-Creb, p-Nrf2, p-Foxo3a). The authors show blots for the phosphorylated forms but not the blot of the unphosphorylated form. This is not usual and, more important, does not allow to say something on the phosphorylation state of that protein. Indeed, the more intense or less intense signal could be the result of higher or lesser amount of protein not of change in the state of phosphorylation of that protein. Given that, statements as “6-OHDA strongly increased…the phosphorylation of JNK protein levels” (page 11, lines 362-363) must be changed.

  1. Results shown in figures 8, 9 and 10 are poorly described. After a brief introduction to all the results shown in these figures (please add figure 10 at line 402, page 15, as it reports data on mitochondrial morphology), the authors say: “6-OHDA significantly decreased mitochondrial markers, while Met and, to a lower extent, Tau were able to counteract mitochondrial dysfunction, mitochondrial ROS formation and restore mitochondrial membrane potential (MMP).” What does it mean decreased mitochondrial markers? When I look at figure 8, I see that Mitosox signal increases in 6-OHDA treated cells. Looking at Figure 9, I read that the approach is TMRM but I do not see here, in the legend or in methods that this approach enables to investigate MMP. In general, the results seem to be written only for expert in this field. Please rewrite statement at page 15, lines 402-405, and explain in detail which are the results shown in each figure.

  1. Figure 10 reports a qualitative analysis of mitochondrial morphology. On this basis, it is hard to state that mitochondrial fragmentation is inhibited (page 18, line 421). Can the authors supply a quantitative analysis of their images or, alternatively, support this statement with the analysis of fission-fusion processes by Western blotting?

  1. Discussion has been written as it was the second part of the introduction. The results are not really discussed and interpreted in perspective of previous studies. Next, the schemes reported in figure 12 and 13 could be useful in a review but not in an original paper. Here, a model explaining the novelty of the work, if any, could be more appropriate.

  1. Figure 2: the green signal in MA plots is not visible.

  1. Please check for abbreviations, punctation, and symbols:

- sometimes hours are abbreviated to h or hr and minutes to min, sometimes they are not abbreviated.

- decimal points are usually reported as full-stops but sometimes are indicated as commas. Please correct the text and figure S1, accordingly.

- replace * with x (page 5, line 211)

-for statistical analyses, p must be replaced by p in both text and legends

  1. Figure 12, legend: “replaces” instead of “replace”.

Round 2

Reviewer 2 Report

The revised version of the paper includes all the suggestions except for one minor point: decimal points remain indicated as commas in figure S1. Following this correction, the manuscript is suitable for publication.

Author Response

We would like to thank again the Reviewer for the valuable comments which helped to improve our research article. We now addressed also the last minor point as suggested (replacing the comma with the dot in Figure S1).